

# Constitution of a comprehensive phytochemical profile and network pharmacology based investigation to decipher molecular mechanisms of *Teucrium polium* L. in the treatment of type 2 diabetes mellitus

Vahap Murat Kutluay[1,*] and Neziha Yagmur Diker[2,*]

[1] Department of Pharmacognosy, Faculty of Pharmacy, Hacettepe University, Ankara, Turkey
[2] Department of Pharmaceutical Botany, Faculty of Pharmacy, Hacettepe University, Ankara, Turkey
[*] These authors contributed equally to this work.

Corresponding author
Vahap Murat Kutluay,
muratkutluay@hacettepe.edu.tr

## ABSTRACT

**Background.** Type 2 diabetes mellitus (T2DM) is a metabolic disease affecting a huge population worldwide. *Teucrium polium* L. has been used as a folk medicine for the treatment of T2DM in Anatolia, Turkey. The antihyperglycemic effect of the plant was reported previously. However, there was no detailed study on the underlying molecular mechanisms. In this study, we generated a research plan to clarify the active constituents of the extract and uncover the molecular mechanisms using network pharmacology analysis.

**Methods.** For this purpose, we composed a dataset of 126 compounds for the phytochemical profile of the aerial parts of *T. polium*. Drug-likeness of the compounds was evaluated, and 52 compounds were selected for further investigation. A total of 252 T2DM related targets hit by selected compounds were subjected to DAVID database.

**Results.** The KEGG pathway analysis showed enrichment for the TNF signaling pathway, insulin resistance, the HIF-1 signaling pathway, apoptosis, the PI3K-AKT signaling pathway, the FOXO signaling pathway, the insulin signaling pathway, and type 2 diabetes mellitus which are related to T2DM . AKT1, IL6, STAT3, TP53, INS, and VEGFA were found to be key targets in protein-protein interaction. Besides these key targets, with this study the role of GSK3β, GLUT4, and PDX1 were also discussed through literature and considered as important targets in the antidiabetic effect of *T. polium*. Various compounds of *T. polium* were shown to interact with the key targets activating PI3K-AKT and insulin signaling pathways.

**Conclusions.** According to these findings, mainly phenolic compounds were identified as the active components and IRS1/PI3K/AKT signaling and insulin resistance were identified as the main pathways regulated by *T. polium*. This study reveals the relationship of the compounds in *T. polium* with the targets of T2DM in human. Our findings suggested the use of *T. polium* as an effective herbal drug in the treatment of T2DM and provides new insights for further research on the antidiabetic effect of *T. polium*.

# INTRODUCTION

Diabetes mellitus is a metabolic disease characterized by high blood glucose levels. According to the reports of WHO, about 422 million people live with diabetes. Diabetes was a direct cause of about 1.6 million deaths only in 2016. Among adults, 90% of the patients have type 2 diabetes mellitus (T2DM) (*Holman, Young & Gadsby, 2015*). In T2DM, β-cell dysfunction and/or insulin resistance results with hyperglycemia and high glucose levels in blood. Patients with T2DM are under risk for some complications that diabetes can cause such as cardiovascular disease, renal disease, diabetic retinopathy and neuropathy (*Zheng, Ley & Hu, 2018*). In T2DM, multitarget treatment is used to overcome the defects caused in the organism (*He et al., 2019*; *Vuylsteke et al., 2015*).

The genus *Teucrium* L., a member of Lamiaceae family (Subfamily Ajugoideaea), has a cosmopolitan distribution and including about 250 species spread worldwide (*Stevens, 2017*). In Turkey, *Teucrium polium* L. is known as ''Acıyavşan'' and used as a traditional medicine for the treatment of diabetes (*Arıtuluk & Ezer, 2012*). In Algeria and Iran, *T. polium* is used traditionally for the treatment of diabetes (*Chinsembu, 2019*; *Rezaei et al., 2015*). The infusion or decoction of aerial parts is used frequently for the treatment of diabetes, stomachache, hemorrhoid, common colds, abdominal pains, antipyretics, and sunstroke as internally (*Tuzlacı, 2016*). *T. polium* has been shown to have mainly flavonoids, phenolic acids, phenylethanoid glycosides, and terpenoids mainly diterpenoids (*Bahramikia & Yazdanparast, 2012*). Phytotherapeutic effects of *T. polium*, such as antioxidant, antimutagenic, cytotoxic, anticancer, hepatoprotective, anti-inflammatory, hypolipidemic, hypoglycemic, antinociceptive, antispasmodic, antiulcer, antibacterial, antiviral, and antifungal activities have been shown by in vivo or in vitro assays (*Bahramikia & Yazdanparast, 2011*; *Hasani-Ranjbar et al., 2010*).

The hypoglycemic effect of *T. polium* was shown by several reports (*Esmaeili & Yazdanparast, 2004*; *Gharaibeh, Elayan & Salhab, 1988*; *Shahraki et al., 2007*; *Yazdanparast, Esmaeili & Helan, 2005*). The hypoglycemic effect of *T. polium* was observed by *Gharaibeh, Elayan & Salhab (1988)* for the first time. In the research, the decoction of aerial parts was tested through three different administrations (oral, intraperitoneal, and intravenous) in normoglycemic and streptozotocin-induced hyperglycemic rats. In all administration ways of *T. polium* decoction, it had caused a decrease in blood glucose concentration. The decoction of *T. polium* had a decrease of 20.5% in blood glucose concentration by oral administration while intraperitoneal and intravenous administration of the decoction had a decrease of 26.5% and 44%, respectively. The study has suggested that the hypoglycemic effect of *T. polium* was a result of an increase in the peripheral utilization of glucose (*Gharaibeh, Elayan & Salhab, 1988*). In another study, administration of ethanol-water (7:3) extract of aerial parts of *T. polium* per six weeks, resulted in a decrease of 64% in blood glucose levels of streptozotocin-induced hyperglycemic rats. The result of this study also had

proved that *T. polium* extract reduced blood glucose concentration by increasing pancreatic insulin secretion dose-dependently (*Esmaeili & Yazdanparast, 2004*). Although there are reports about the hypoglycemic effect of *T. polium*, there is still a lack of information for underlying mechanisms. In a recent study, to elucidate molecular mechanisms, effects of *T. polium* extract on pancreatic islets cells regeneration was investigated. It was found that the antidiabetic effect of *T. polium* was connected with the antioxidant defense system and Pdx1 expression in the JNK pathway (*Tabatabaie & Yazdanparast, 2017*).

Plants have been used for the treatment of various diseases in folk medicine. These herbal preparations consist of multiple compounds that target multiple proteins in an organism. This suits well with the multicomponent-multitarget paradigm (*Zhang et al., 2019*). Network pharmacology provides information to understand the underlying mechanisms of therapeutic and adverse effects of these multicomponent therapeutics (*Hopkins, 2008*; *Keith, Borisy & Stockwell, 2005*; *Li & Zhang, 2013*). Unlike the trend in drug research studies held in the 20th century which aims for single components affecting single targets, nowadays researchers focus on multicomponent therapeutics. Network pharmacology is a rising trend in the 21st century, mainly after 2010 (*Lu et al., 2019*). Network pharmacology studies revealed molecular mechanisms of several Traditional Chinese Medicine (TCM) recipes in the treatment of complex diseases already (*Chen et al., 2019*; *Chen et al., 2018*; *Xiang et al., 2019*).

In this study, the underlying mechanisms of *T. polium* in the treatment of diabetes were aimed to be elucidated. For this purpose, firstly the phytochemical content of the plant was screened through a detailed literature search. Compounds reported from *T. polium* were selected based on their drug-likeness properties and screened for their potential targets that play a role in the biological processes. Therapeutical targets of T2DM and targets of the compounds were merged for further investigations. Protein-protein interaction (PPI) network of common targets was constructed. Key targets were determined and the role of targets in T2DM pathways was discussed.

# MATERIALS & METHODS

## Literature based search for phytochemical content of *T. polium*
Previous phytochemical studies on *T. polium* were reviewed and compounds that were isolated or determined listed. Literature search was performed using 'Scopus' and 'Web of Science–Clarivate' databases with the keyword '*Teucrium polium*' upto June 2020. After the review process, it was found that 126 compounds were reported from *T. polium* by several reports. Due to their structure, compounds were listed under 3 groups (phenolics, terpenoids and amino acid derivatives). All the compounds were converted to Canonical SMILES format using PubChem (https://pubchem.ncbi.nlm.nih.gov/) or CS Chemdraw Ultra.

## Evaluation of drug likeness of the compounds
The absorption and permeation abilities of the compounds in the extract play a critical role in the biological activity observed. In this study, Lipinski's rule of five was used to filter compounds which possess good absorption and permeation so that could be

a new drug candidate. According to this criteria compounds which have; (i) molecular weight (MW) greater than 500, (ii) the calculated logP value above 5, (iii) more than 5 hydrogen bond donors (HBD) and (iv) more than 10 hydrogen bond acceptors (HBA) were filtered (*Lipinski et al., 2001*; *Turner & Agatonovic-Kustrin, 2007*). All 126 compounds were subjected to SWISSADME to obtain data for pharmacokinetics and drug-likeness (*Daina, Michielin & Zoete, 2017*). 52 compounds meet the Lipinski's rule of 5′and further used to construct 'Compound-Target' network. The detailed results for all the dataset of 126 compounds can be found in Table S1.

## Construction of 'Compound-Target' network

In an herbal extract, each compound has an interaction with specific targets. The biological effects of the extract are a result of these interactions. Targets of the selected 52 compounds were searched through TCSMP (Traditional Chinese Medicine Systems Pharmacology Database and Analysis Platform), and SYMMAP databases (*Ru et al., 2014*; *Wu et al., 2018*). Compounds were also subjected to SwissTargetPrediction for target fishing (*Daina, Michielin & Zoete, 2019*). During these screening, targets were limited for *Homo sapiens*. To avoid confusion, targets obtained were screened for UniprotKB ID for their unique identifiers (http://www.uniprot.org/) (*UniProt Consortium, 2018*). Duplicate targets for the same compounds were removed and 'Compound-Target' network was obtained including 704 targets (Table S2). Cytoscape 3.8.0 was used for the visualization of the network (*Shannon et al., 2003*).

## Collection of T2DM targets and network construction

T2DM related genes were collected from DisGeNET (https://www.disgenet.org/). 'Type 2 diabetes mellitus' was used as a keyword. DisGeNET is a platform with collections of genes associated with diseases (*Bauer-Mehren et al., 2010*). 1513 genes related to T2DM were obtained. The data obtained were transferred to Cytoscape 3.8.0.

## 'Compound-Target-Disease' network and the 'Protein-Protein Interaction' network construction

For further investigation, the intersection of 'Compound-Target' network and T2DM related genes were set as 'Compound-Target-Disease' network. This network consisted of 252 genes (Table S3). For an illustration of the roles of selected genes in biological systems, the STRING database (http://string-db.org/, version 11) was used. STRING is a database that helps understanding associations between expressed proteins in a cellular function (*Szklarczyk et al., 2018*). Protein-protein interaction (PPI) map of 252 genes were generated. The confidence score was set as high ($> 0.7$). The key targets were defined using topological analysis. Topological network parameters cover some properties such as; degree distributions, stress centrality, betweenness centrality, closeness centrality (*Doncheva et al., 2012*). In this study, degree distributions were selected to identify key targets.

## Gene enrichment analysis

DAVID Bioinformatics Resources 6.8 was used in gene enrichment analysis. DAVID database integrates biological knowledge with analytical tools that provide bioinformatic

annotations (*Huang, Sherman & Lempicki, 2008*; *Huang, Sherman & Lempicki, 2009*). The 252 genes which were common for compounds and disease were uploaded to DAVID (https://david.ncifcrf.gov/). The results were listed based on their *p* values. Top 20 results with lower *p* value were selected. The results of gene ontology (GO) function and (Kyoto Encyclopedia of Genes and Genomes) KEGG pathway analysis were evaluated and discussed (*Ashburner et al., 2000*; *Gene Ontology Consortium, 2019*; *Mi et al., 2019*).

### Construction of 'Compound-Target-Pathway' network

'Compound-Target-Disease' network and selected KEGG pathways were intersected to give 'Compound-Target-Pathway' network. Cytoscape was used for visualization.

## RESULTS

### Screening of chemical compounds in *T. polium* and selection for the potential active compounds

Through a detailed literature search, 126 compounds were listed in the aerial parts of *T. polium* (Table S4). Mainly phenolic compounds (flavonoids, phenylethanoid glycosides, phenolic acids), terpenoids (secoiridoids, iridoids, sesquiterpenoids, diterpenoids, triterpenoids) and amino acid derivatives (cyanogenic glycosides) were identified. Drug-likeness of these compounds were scanned through Lipinski's rule of 5 (the parameters of the compounds were given in Table S1). 80 compounds that met the selected criteria were searched for their potential targets using SYMMAP, TCMSP and Swiss Target Prediction databases. The databases provided information for 52 compounds (Table 1).

### Analysis of compound-target interactions and determination of the common targets of *T. polium* and T2DM related genes

There is a total of 704 genes related to 52 compounds. The 'Compound-Target' network consists of 756 nodes and 4023 edges. Quercetin, apigenin and luteolin were found to be in relation with more targets than the other compounds (edge numbers were 254, 184 and 158 respectively). For further investigation, the common targets for *T. polium* and T2DM were determined. Firstly, 1513 T2DM related genes were imported from DisGeNet. All the targets were converted to Uniprot IDs to avoid confusion. The merge process of compound-target network and disease-target network resulted in 252 common targets (Fig. 1). These targets were selected for further investigation to understand the mechanisms of *T. polium* in the treatment of T2DM.

### Construction of PPI networks and determination of the key targets

To understand the metabolic processes, PPIs play a key role. It comprises a network including direct and indirect interactions between proteins which give researchers new insights in understanding biological phenomena (*Ijaz, Ansari & Iqbal, 2018*; *Szklarczyk et al., 2018*). To clarify the key targets in the 'Compound-Target-Disease' network, the target genes were subjected to STRING 11.0 using a confidence score of > 0.7 (high) to achieve PPI network. The PPI network had 252 nodes and 1912 edges. According to topological analysis, degree distributions were evaluated. Degree shows the interaction numbers of the targets within the network. The nodes with a higher degree are referred to as a hub. The

**Table 1  The molecular formula and structures of the 52 compounds that used in network pharmacology analysis.**

| No | Compound code | Compound name | Formula | Structure |
|---|---|---|---|---|
| 1 | TP1 | 4′,7-dimethoxy apigenin | $C_{17}H_{14}O_5$ | |
| 2 | TP2 | 4′-O-methyl luteolin | $C_{16}H_{12}O_6$ | |
| 3 | TP3 | 6-hydroxy luteolin | $C_{15}H_{10}O_7$ | |
| 4 | TP4 | Acacetin | $C_{16}H_{12}O_5$ | |
| 5 | TP5 | Apigenin | $C_{15}H_{10}O_5$ | |
| 6 | TP6 | Catechin | $C_{15}H_{14}O_6$ | |
| 7 | TP7 | Cirsilineol | $C_{18}H_{16}O_7$ | |
| 8 | TP8 | Cirsiliol | $C_{17}H_{14}O_7$ | |
| 9 | TP9 | Cirsimaritin | $C_{17}H_{14}O_6$ | |
| 10 | TP10 | Eupatorin | $C_{18}H_{16}O_7$ | |

**Table 1** (*continued*)

| No | Compound code | Compound name | Formula | Structure |
|----|---------------|---------------|---------|-----------|
| 11 | TP11 | Isoscutellarein | $C_{15}H_{10}O_6$ | |
| 12 | TP12 | Jaceosidin | $C_{17}H_{14}O_7$ | |
| 13 | TP13 | Luteolin | $C_{15}H_{10}O_6$ | |
| 14 | TP14 | Quercetin | $C_{15}H_{10}O_7$ | |
| 15 | TP15 | Caffeic acid | $C_9H_8O_4$ | |
| 16 | TP16 | Gallic acid | $C_7H_6O_5$ | |
| 17 | TP17 | *p*-Coumaric acid | $C_9H_8O_3$ | |
| 18 | TP18 | *t*-Ferulic acid | $C_{10}H_{10}O_4$ | |
| 19 | TP19 | Vanillic acid | $C_8H_8O_4$ | |
| 20 | TP20 | 2-(3,4-dihydroxyphenyl)ethanol | $C_8H_{10}O_3$ | |

**Table 1** (*continued*)

| No | Compound code | Compound name | Formula | Structure |
|----|---------------|---------------|---------|-----------|
| 21 | TP21 | Tyrosol | $C_8H_{10}O_2$ | |
| 22 | TP22 | 4α-[(β-D-glucopyranosyloxy)methyl]-5α-(2-hydroxyethyl)-3-methylcyclopent-2-en-1-one | $C_{15}H_{24}O_8$ | |
| 23 | TP23 | 5α-[2-(β-D-glucopyranosyloxy)ethyl]-4α-hydroxymethyl-3-methylcyclopent-2-en-1-one | $C_{15}H_{24}O_8$ | |
| 24 | TP24 | 20-O-acetyl-teucrasiatin | $C_{24}H_{30}O_8$ | |
| 25 | TP25 | Capitatin | $C_{24}H_{28}O_9$ | |
| 26 | TP26 | Clerodane-6,7-dione | $C_{22}H_{24}O_8$ | |
| 27 | TP27 | Teubutilin A | $C_{22}H_{28}O_6$ | |
| 28 | TP28 | Teupolin VII | $C_{20}H_{26}O_5$ | |
| 29 | TP29 | Teupolin VIII | $C_{19}H_{24}O_5$ | |

**Table 1** (*continued*)

| No | Compound code | Compound name | Formula | Structure |
|----|---------------|---------------|---------|-----------|
| 30 | TP30 | (1R, 4S, 10R) 10,11-dimethyl-dicyclohex-5(6)-en-1,4-diol-7-one | $C_{12}H_{18}O_3$ | |
| 31 | TP31 | (1R,6R,7R,8S,11R)-1,6-dihydroxy-4,11-dimethyl-germacran-4(5), 10(14)-dien-8,12-olide | $C_{15}H_{22}O_4$ | |
| 32 | TP32 | (10R,1R,4S,5S,6R,7S)-4,10-die-poxygermacran-6-ol | $C_{15}H_{26}O_3$ | |
| 33 | TP33 | Prunasin | $C_{14}H_{17}NO_6$ | |
| 34 | TP34 | 1$\alpha$-hydroxy isoondetamnone | $C_{12}H_{18}O_2$ | |
| 35 | TP35 | 4$\beta$,5$\alpha$-Epoxy-7$\alpha$H-germacr-10(14)-en-6$\beta$-ol-1-one | $C_{15}H_{24}O_3$ | |
| 36 | TP36 | 4$\beta$,5$\alpha$-Epoxy-7$\alpha$H-germacr-10(14)-en,1$\beta$-hydroperoxyl,6$\beta$-ol | $C_{15}H_{26}O_4$ | |
| 37 | TP37 | 4$\beta$,5$\beta$-Epoxy-7$\alpha$H-germacr-10(14)-en,1$\beta$-hydroperoxyl,6$\beta$-ol | $C_{15}H_{26}O_4$ | |
| 38 | TP38 | 4$\alpha$,5$\beta$-epoxy-7$\alpha$H-germacr-10(14)-en,1$\beta$-hydroperoxyl,6$\alpha$-ol | $C_{15}H_{26}O_4$ | |

**Table 1** (*continued*)

| No | Compound code | Compound name | Formula | Structure |
|----|---------------|---------------|---------|-----------|
| 39 | TP39 | 10α,1β;4β,5α-diepoxy-7αH-germacrm-6-ol | $C_{15}H_{26}O_3$ | |
| 40 | TP40 | Teucladiol | $C_{15}H_{26}O_2$ | |
| 41 | TP41 | 4β,6β-dihydroxy-1α,5β(H)-guai-9-ene | $C_{15}H_{26}O_2$ | |
| 42 | TP42 | Oplopanone | $C_{15}H_{26}O_2$ | |
| 43 | TP43 | Oxyphyllenodiol A | $C_{14}H_{22}O_3$ | |
| 44 | TP44 | Arteincultone | $C_{15}H_{24}O_4$ | |
| 45 | TP45 | Ladanein | $C_{17}H_{14}O_6$ | |
| 46 | TP46 | Salvigenin | $C_{18}H_{16}O_6$ | |
| 47 | TP47 | 5,3′,4′-trihydroxy-3,7-dimethoxyflavone | $C_{17}H_{14}O_7$ | |

**Table 1** (*continued*)

| No | Compound code | Compound name | Formula | Structure |
|----|---------------|---------------|---------|-----------|
| 48 | TP48 | Jaranol | $C_{17}H_{14}O_6$ | |
| 49 | TP49 | $\beta$-eudesmol | $C_{15}H_{26}O$ | |
| 50 | TP50 | $\alpha$-Cadinol | $C_{15}H_{26}O$ | |
| 51 | TP51 | 7-epi-Eudesm-4(15)-ene-l$\beta$,6$\alpha$-diol | $C_{15}H_{26}O_2$ | |
| 52 | TP52 | 7-epi-Eudesm-4(15)-ene-l$\beta$,6$\beta$-diol | $C_{15}H_{26}O_2$ | |

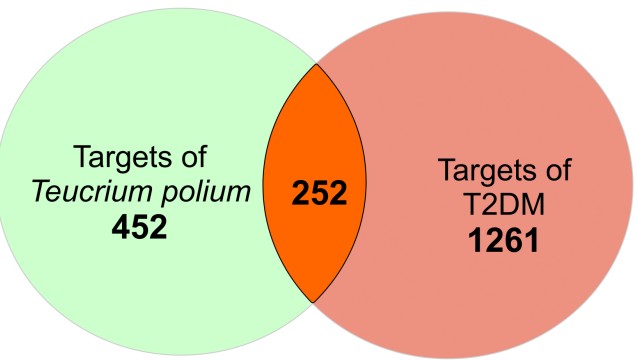

**Figure 1** **The scheme for intersection of *T. polium* and type 2 diabetes mellitus (T2DM) targets.**

hub plays a key role in the biological process as it is related with more targets. Two-fold of the mean of the degree was selected as a threshold for the determination of key targets. 37 targets with a higher degree than 30.3 was thought to have a critical role for the mechanism of action (AKT1, INS, VEGFA, IL6, TP53, STAT3, MAPK1, APP, TNF, MAPK8, CXCL8, EGFR, PIK3CA, PIK3R1, SRC, MMP9, IL10, PTGS2, IL1B, CCL2, RELA, HRAS, GAPDH, PTEN, IL2, IL4, MTOR, TLR4, CASP3, JAK2, ICAM1, ESR1, FGF2, CXCL10, PPARG, MMP2, MAPK14). Acacetin (TP4), apigenin (TP5), jaceosidin (TP12), luteolin (TP13), quercetin (TP14), and caffeic acid (TP15) showed higher interactions with the key targets

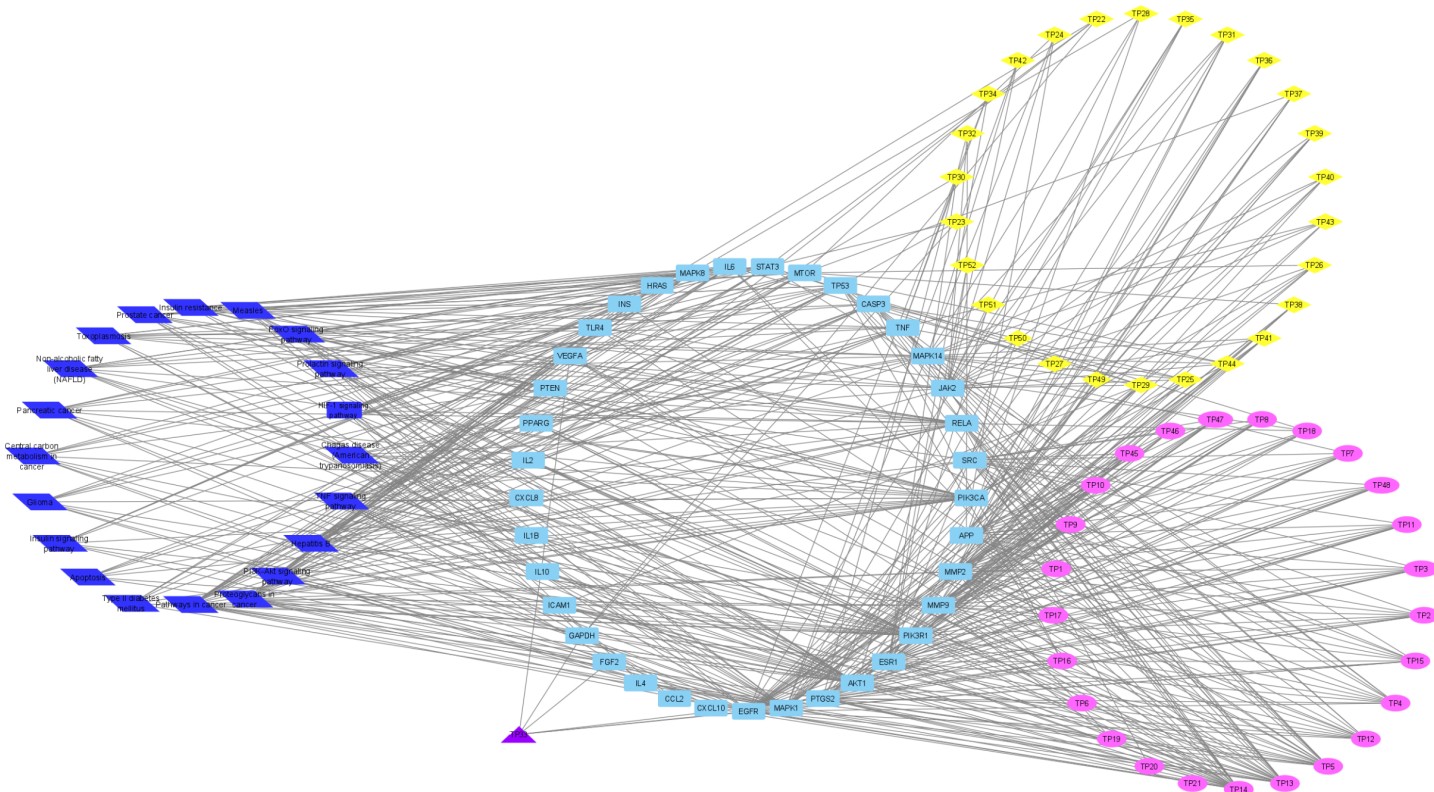

**Figure 2** **'Compound-Key Target-Pathway' network of *T. polium*.** Pink nodes represent phenolic compounds, yellow nodes represent terpenoids and purple node represent cyanogenic glycoside. Key targets are given in light blue nodes and pathways are given in dark blue nodes.

and might have a role in the antidiabetic effects of *T. polium* (Fig. 2). PPI network for key targets had 37 nodes and 425 edges. Interleukin-6 (IL6), signal transducer and activator of transcription 3 (STAT3), mitogen-activated protein kinase 1 (MAPK1), insulin (INS), and vascular endothelial growth factor A (VEGFA) were the proteins with a higher number of interactions (Fig. 3).

## Gene enrichment analysis using DAVID database for GO and KEGG

252 common targets were subjected to DAVID database for gene enrichments. GO and KEGG gene enrichment results were put in order according to their *p* values. GO enrichment results were given in three parts; molecular, biological, and cellular. Top 20 results for each analysis were plotted in a graph produced by Graphpad Prism 6 (Figs. 4 and 5).

The results for GO analysis were evaluated through related terms option of DAVID database. According to biological process results; response to drug (GO:0042493), negative regulation of apoptotic process (GO:0043066) and positive regulation of transcription from RNA polymerase II promoter (GO:0045944) showed higher target numbers in count (Fig. 4). Negative regulation of apoptotic process, inflammatory response (GO:0006954), positive regulation of cell proliferation (GO:0008284), glucose homeostasis (GO:0042593) and glucose transport (GO:0015758) were found to be related with at least one of the

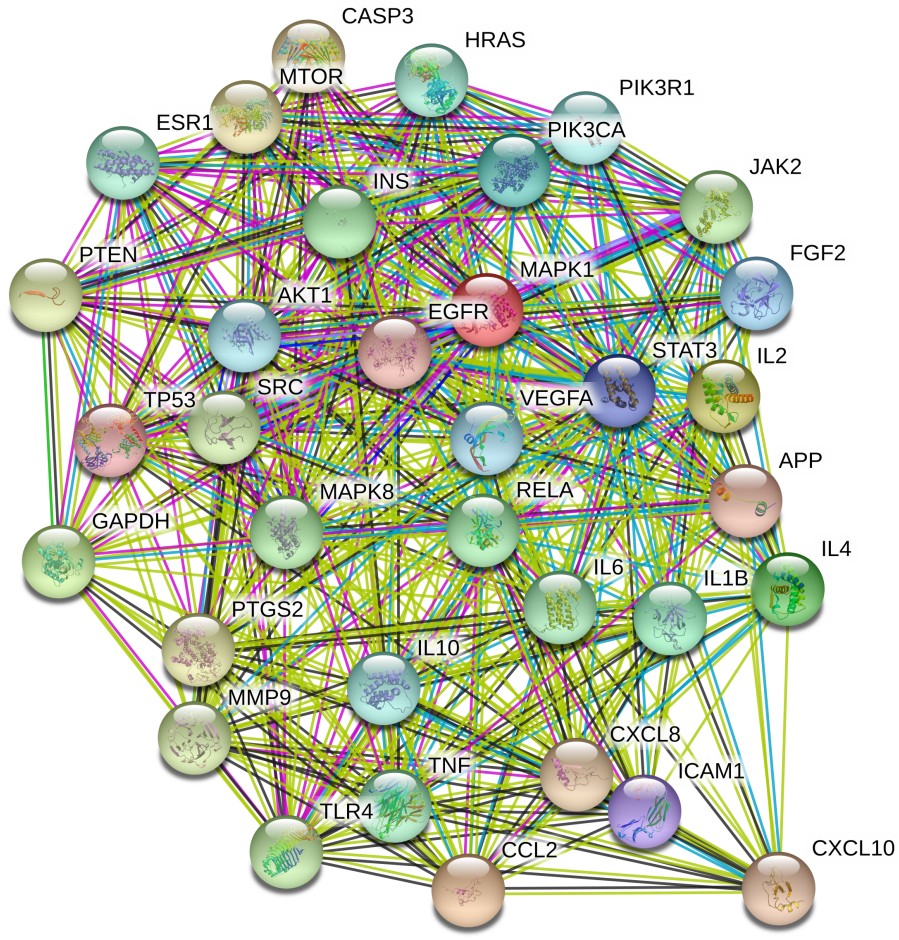

**Figure 3** **The PPI network of the 37 key targets obtained from STRING v11.**

KEGG pathways such as PI3K-Akt signaling pathway (hsa04151), TNF signaling pathway (hsa04668), insulin resistance (hsa04931), insulin signaling pathway (hsa04910), FoxO signaling pathway (hsa04068), adipocytokine signaling pathway (hsa04920), AMPK signaling pathway (hsa04152) and type 2 diabetes mellitus (hsa04930). Molecular function results with higher target numbers were protein binding (GO:0005515), protein homodimerization activity (GO:0042803), and identical protein binding (GO:0042802) (Fig. 4). Kinase activity (GO:0016301) and insulin receptor substrate binding (GO:0043560) were found to be related to at least one of the KEGG pathways such as insulin resistance, insulin signaling pathway, FoxO signaling pathway, PI3K-Akt signaling pathway, and type 2 diabetes mellitus.

KEGG enrichment results supported these findings. Results of 252 common targets were listed as (related with T2DM); TNF signaling pathway, insulin resistance, apoptosis, HIF-1 signaling pathway, PI3K-Akt signaling pathway, FoxO signaling pathway, insulin signaling pathway and type 2 diabetes mellitus (Fig. 5). The top 20 results according to the *p* values

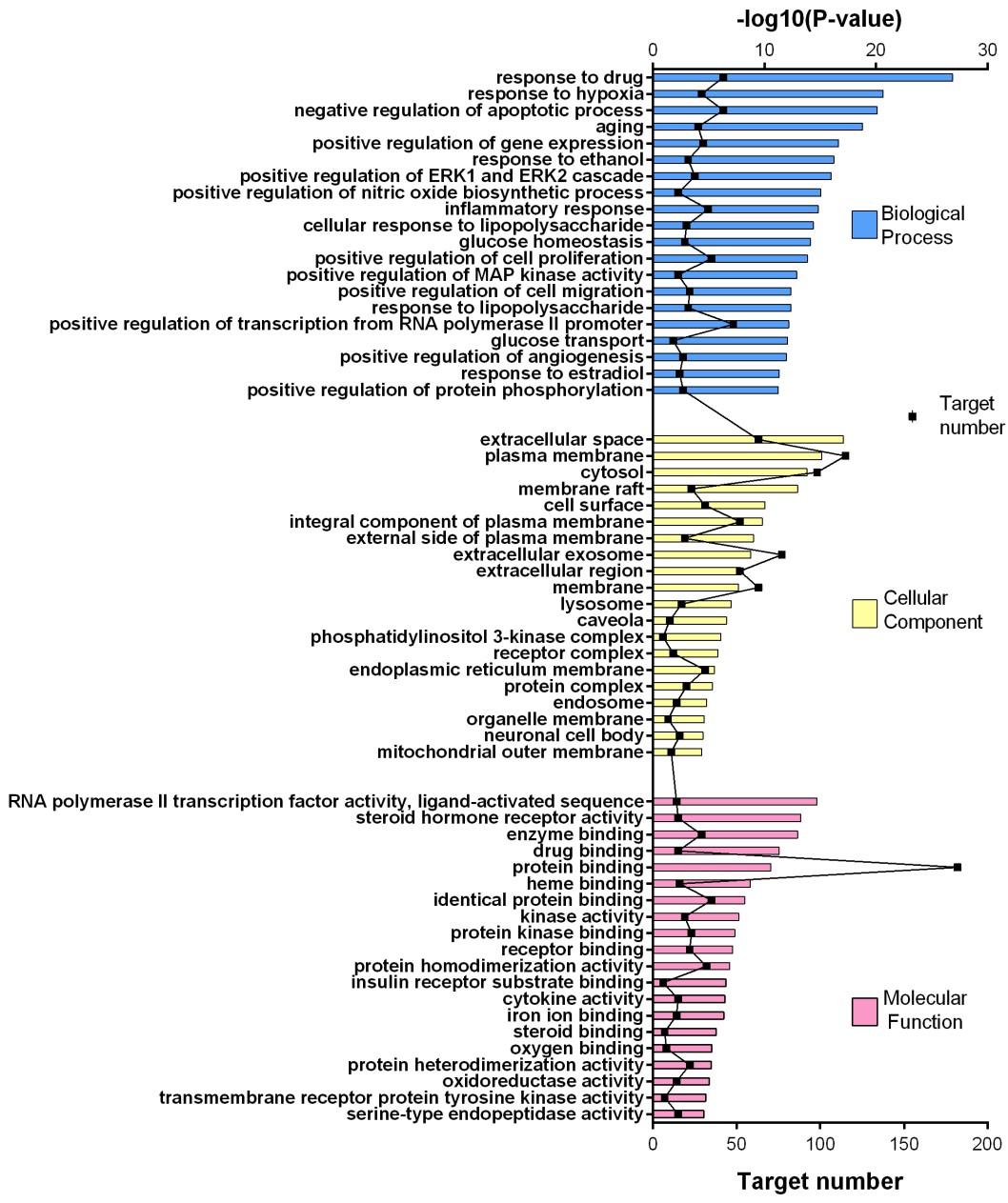

**Figure 4  GO gene enrichment analysis for 252 common targets (top 20 results according to the *p* value).**

suggested that, compounds reported from *T. polium* might also leads new insights for the treatment of cancer (Table 2).

## DISCUSSION

The potent hypoglycemic effect of *T. polium* extract has been reported by several reports (*Esmaeili & Yazdanparast, 2004*; *Gharaibeh, Elayan & Salhab, 1988*; *Tabatabaie*
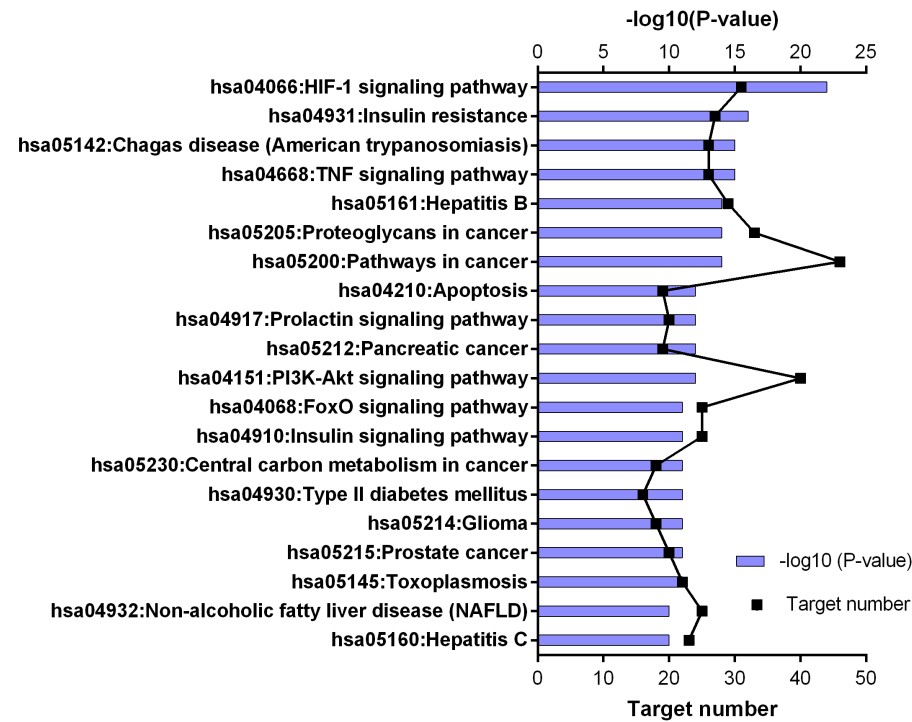

**Figure 5   KEGG pathway enrichment results for 252 candidate target genes (top 20 results according to the _p_ value).**

_& Yazdanparast, 2017_; _Yazdanparast, Esmaeili & Helan, 2005_). According to the study performed by Tabatabaie and Yazdanparast, _T. polium_ extract lowered fasting blood glucose levels closely to the control group. In addition to the hypoglycemic effect, _T. polium_ treated rats had lower triglyceride and cholesterol levels when compared with diabetic rats (_Tabatabaie & Yazdanparast, 2017_). These experimental data show the potential of _T. polium_ as a promising herb in the treatment of T2DM.

According to the findings of this study, we are considering that _T. polium_ show its antidiabetic effect via enhancing β-cell number and function and exhibiting insulin-like effect through PI3K-AKT pathway. Previous reports performed on the extract and compounds of _T. polium_ support our findings of the network pharmacology assisted analysis. According to the KEGG enrichment results in our study, 18 key targets take part in PI3K-AKT pathway and 11 key targets take part in insulin resistance pathway (Table 2). For this view, it is essential to understand the roles of these pathways in T2DM.

T2DM is appeared owing to two fundamental defects that are insulin resistance and impaired β-cell function which are caused by long-term hyperglycemia (_Cheatham & Kahn, 1995_). Insulin resistance is characterized as reduced insulin sensitivity in the target tissue (like skeletal muscle, liver, and adipose). It is connected with the pathogenesis of metabolic diseases like obesity, type 2 diabetes, hypertension, cardiovascular diseases, and fatty liver disease (_Draznin, 2020_).

**Table 2  KEGG pathway enrichment results for 252 common targets together with the key targets involved.**

| No | Pathway name | Key target | Nodes |
|---|---|---|---|
| 1 | HIF-1 signaling pathway | AKT1, RELA, EGFR, GAPDH, INS, IL6, MTOR, MAPK1, PIK3CA, PIK3R1, STAT3, TLR4, VEGFA | 13 |
| 2 | Insulin resistance | AKT1, RELA, INS, IL6, MTOR, MAPK8, PTEN, PIK3CA, PIK3R1, STAT3, TNF | 11 |
| 3 | Chagas disease (American trypanosomiasis) | AKT1, CCL2, CXCL8, RELA, IL1B, IL10, IL2, IL6, MAPK1, MAPK14, MAPK8, PIK3CA, PIK3R1, TLR4, TNF | 15 |
| 4 | TNF signaling pathway | AKT1, CCL2, CXCL10, RELA, CASP3, ICAM1, IL1B, IL6, MMP9, MAPK1, MAPK14, MAPK8, PIK3CA, PIK3R1, PTGS2, TNF | 16 |
| 5 | Hepatitis B | AKT1, CXCL8, HRAS, RELA, SRC, CASP3, IL6, MMP9, MAPK1, MAPK8, PTEN, PIK3CA, PIK3R1, STAT3, TLR4, TNF, TP53 | 17 |
| 6 | Proteoglycans in cancer | AKT1, HRAS, SRC, CASP3, EGFR, ESR1, FGF2, MMP2, MMP9, MTOR, MAPK1, MAPK14, PIK3CA, PIK3R1, STAT3, TLR4, TNF, TP53, VEGFA | 19 |
| 7 | Pathways in cancer | AKT1, CXCL8, HRAS, RELA, CASP3, EGFR, FGF2, IL6, MMP2, MMP9, MTOR, MAPK1, MAPK8, PPARG, PTEN, PIK3CA, PIK3R1, PTG2, STAT3, TP53, VEGFA | 21 |
| 8 | Apoptosis | AKT1, RELA, CASP3, PIK3CA, PIK3R1, TNF, TP53 | 7 |
| 9 | Prolactin signaling pathway | AKT1, HRAS, JAK2, RELA, SRC, ESR1, INS, MAPK1, MAPK14, MAPK8, PIK3CA, PIK3R1, STAT3 | 13 |
| 10 | Pancreatic cancer | AKT1, RELA, EGFR, MAPK1, MAPK8, PIK3CA, PIK3R1, STAT3, TP53, VEGFA | 10 |
| 11 | PI3K-Akt signaling pathway | AKT1, HRAS, JAK2, RELA, EGFR, EGFR, INS, IL2, IL4, IL6, MTOR, MAPK1, PTEN, PIK3CA, PIK3R1, TLR4, TP53, VEGFA | 18 |
| 12 | FoxO signaling pathway | AKT1, HRAS, EGFR, INS, IL10, IL6, MAPK1, MAPK14, MAPK8, PTEN, PIK3CA, PIK3R1, STAT3 | 13 |
| 13 | Insulin signaling pathway | AKT1, HRAS, INS, MTOR, MAPK1, MAPK8, PIK3CA, PIK3R1 | 8 |
| 14 | Central carbon metabolism in cancer | AKT1, HRAS, EGFR, MTOR, MAPK1, PTEN, PIK3CA, PIK3R1, TP53 | 9 |
| 15 | Type II diabetes mellitus | INS, MTOR, MAPK1, MAPK8, PIK3CA, PIK3R1, TNF | 7 |
| 16 | Glioma | AKT1, HRAS, EGFR, MTOR, MAPK1, PTEN, PIK3CA, PIK3R1, TP53 | 9 |
| 17 | Prostate cancer | AKT1, HRAS, RELA, EGFR, INS, MTOR, MAPK1, PTEN, PIK3CA, PIK3R1, TP53 | 11 |
| 18 | Toxoplasmosis | AKT1, JAK2, RELA, CASP3, IL10, MAPK1, MAPK14, MAPK8, STAT3, TLR4, TNF | 11 |
| 19 | Non-alcoholic fatty liver disease (NAFLD) | AKT1, CXCL8, RELA, CASP3, INS, IL1B, IL6, MAPK8, PIK3CA, PIK3R1, TNF | 11 |
| 20 | Measles | AKT1, JAK2, RELA, IL1B, IL2, IL4, IL6, PIK3CA, PIK3R1, STAT3, TLR4, TP53 | 12 |

INS, one of the hub genes in the PPI network in our study, regulates glucose metabolism and ensure metabolic homeostasis. It also promotes glycogen synthesis, lipid metabolism, protein synthesis and degradations, gene transcriptions, etc. (*Cheatham & Kahn, 1995*). Additionally, it is an important regulator of pancreatic β-cells growth and proliferation

through the phosphatidylinositol 3-kinase (PI3K)/AKT (also known as protein kinase B-PKB) pathway (*Fujimoto & Polonsky, 2009*).

Insulin receptor activation through insulin binding stimulates PI3K-AKT signaling pathway. AKT is determined as one of the key targets in our study and 4′-*O*-methyl luteolin (TP2), 6-hydroxy luteolin (TP3), apigenin (TP5), isoscutellarein (TP11), jaceosidin (TP12), luteolin (TP13), quercetin (TP14), 20-*O*-acetyl teucrasiatin (TP24), and capitatin (TP25), 4β,5α-epoxy-7 αH-germacr-10(14)-en,1β-hydroperoxyl,6β-ol (TP36), 4α,5β-epoxy-7αH-germacr-10(14)-en,1β-hydroperoxyl,6α-ol (TP38), 5,3′,4′- trihydroxy-3,7-dimethoxyflavone (TP47), jaranol (TP48) showed interaction with the target AKT1 that could have a role in the antidiabetic effect of *T. polium* (Fig. 2). AKT activation, promotes cell survival, proliferation, and growth by controlling key signaling nodes such as glycogen synthase kinase 3 (GSK3), Forkhead Box O (FoxO) transcriptions factors, tuberous sclerosis complex 2 (TSC2) and mechanistic target of rapamycin (mTOR) complex 1 (mTORC1) (*Manning & Toker, 2017*).

In a previous study about molecular mechanisms of the effects of *T. polium* extract on pancreatic β-cells regeneration, it was showed that c-jun N-terminal kinase (JNK) pathway provoked with oxidative stress leads to the inactivation of pancreas/duodenum homeobox protein 1 (PDX1) via FoxO1. In the same study, *T. polium* extract increased FoxO1 phosphorylation and, promoted the expression of PDX1 (*Tabatabaie & Yazdanparast, 2017*). Tyrosol (TP21), a polyphenol reported from *T. polium*, also was shown to inhibit ER-stress induced β-cell apoptosis by JNK phosphorylation (*Lee et al., 2016*). JNK pathway induces insulin receptor substrate 1 (IRS1) inhibition by causing serine phosphorylation which is an important step in downstream of insulin receptor signaling. This impairs the insulin signaling pathway by inhibiting IRS1-IRS2/PI3K-AKT pathway (*Kaneto et al., 2005*). IRS1-IRS2/PI3K-AKT pathway inactivates FoxO1 which supports pancreatic β-cells proliferation by enhancing PDX1 expression (*Kitamura et al., 2002*).

PDX1 has a very important role in pancreatic β-cell function and survival, is regulated through FoxO1 and glycogen synthase kinase 3 beta (GSK3β) (*Fujimoto & Polonsky, 2009*). In skeletal muscle, AKT inactivates GSK3β by phosphorylation that results in a reduction in the phosphorylation of a few GSK3 substrates such as Glycogen Synthase (GS) (*Hermida, Dinesh Kumar & Leslie, 2017*). In pancreatic β-cells of the islets, GSK3β inhibited by AKT, does not phosphorylate PDX1. PDX1 is a critical regulator of pancreatic development and activates glucose transporter 2 (GLUT2), INS, and glucokinase genes (*Humphrey et al., 2010*). Thus, pharmacological inhibition of GSK3β could be substantial in type 2 diabetes treatment (*Sacco et al., 2019*). Even GSK3β has not been found as a key target in our PPI network, a large group of compounds reported in *T. polium* interact with GSK3β, listed as, 4′,7-dimethoxy apigenin (TP1), 4′-*O*-methyl luteolin (TP2), 6-hydroxy luteolin (TP3), acacetin (TP4), apigenin (TP5), cirsilineol (TP7), cirsiliol (TP8), cirsimaritin (TP9), eupatorin (TP10), isoscutellarein (TP11), jaceosidin (TP12), luteolin (TP13), quercetin (TP14), 20-*O*-acetyl teucrasiatin (TP24), capitatin (TP25), clerodane-6,7-dione (TP26), (1R,6R,7R,8S,11R)-1,6-dihydroxy-4,11-dimethyl-germacran-4(5), 10(14)-dien-8,12-olide (TP31), prunasin (TP33), teucladiol (TP40), 4β,6β-dihydroxy-1α,5β(H)-guai-9-ene

(TP41), oplopanone (TP42), ladanein (TP45), salvigenin (TP46), 5,3′,4′- trihydroxy-3,7-dimethoxyflavone (TP47), jaranol (TP48) (Table S5). Both phenolic compounds, terpenoids and a cyanogenic glycoside found to interact with GSK3β.

Furthermore, glucose transporter 4 (GLUT4) induction through insulin-stimulated PI3K/AKT has an important role in whole-body glucose homeostasis by glucose intake at adipose tissue, cardiomyocytes and skeletal muscle cells (*Klip, McGraw & James, 2019*). Previous reports on flavonoids like apigenin (TP5), quercetin (TP14), kaempferol showed enhancement in GLUT4 translocation (*Alkhalidy et al., 2015*; *Hossain et al., 2014*; *Jiang et al., 2019*). According to the findings of our study, apigenin (TP5), luteolin (TP13), and quercetin (TP14) showed interaction with the target GLUT4 (Table S5). By *Kadan et al. (2018)*, it was found that extract of *T. polium* increased translocation of GLUT4 in L6 muscle cells in the absence and presence of insulin when compared with the control. These findings indicated that insulin-like activity of *T. polium* is a result of increase in GLUT4 translocation (*Kadan et al., 2018*). Quercetin (TP14) and gallic acid (TP16) increased glucose uptake through IRS1/PI3K/AKT signaling (*Gandhi et al., 2014*; *Jiang et al., 2019*). Furthermore, it was shown that gallic acid and *p*-coumaric acid (TP17) ameliorate insulin shortage and insulin resistance (*Abdel-Moneim et al., 2018*). A previous study on rats showed that vanillic acid (TP19) upregulates hepatic insulin signaling, insulin receptor, phosphatidylinositol-3 kinase, glucose transporter 2, and phosphorylated acetyl CoA carboxylase expression (*Chang et al., 2015*). Kaempferol treatment also showed to improve β-cell mass in diabetic mice (*Alkhalidy et al., 2015*). Similarly, cirsimaritin (TP9) suppresses apoptosis in β-cells (*Lee et al., 2017*).

IL6, another hub gene in the PPI network in this study, is a proinflammatory cytokine that has a complex role in T2DM. In this study, luteolin (TP13), quercetin (TP14) and teupolin VIII (TP29) showed interaction with the target IL6 (Table S5). Several studies showed that chronic inflammation plays a role in T2DM (*Lehrskov & Christensen, 2019*). Studies on the patients with T2DM showed an increase in IL6 levels in the plasma (*Akbari & Hassan-Zadeh, 2018*). The biological role of IL6 depends on the signaling pathway (*Akbari & Hassan-Zadeh, 2018*). Although there are different opinions on the effects of IL6 in T2DM, recent studies showed that the absence of IL6 resulted in hyperglycemia and higher fat levels in people with obesity (*Kurauti et al., 2017*; *Lehrskov & Christensen, 2019*). However, IL6 also increases the expression of insulin degrading enzyme which is important in glucose metabolism. This enzyme also degrades amyloid β. For this view, IL6 is an important cytokine both in two closely related diseases Alzheimer's disease and T2DM (*Kurauti et al., 2017*). STAT3 activation also negatively regulates another common target GSK3β of Alzheimer's disease and T2DM (*Moh et al., 2008*). STAT3 has a role in cell differentiation in various systems including immune and endocrine systems. However recent studies showed that STAT3 suppression together with Pdx1 expression increased the number of β-cells (*Miura et al., 2018*). Caffeic acid (TP15), *t*-ferulic acid (TP18), 4α-[(β-D-glucopyranosyloxy)methyl]-5α-(2-hydroxyethyl)-3-methylcyclopent-2-en-1-one (TP22), 5α-[2-(β-D-glucopyranosyloxy)ethyl]-4α-hydroxymethyl-3-methylcyclopent-2-en-1-one (TP23), and teupolin VIII (TP29) showed interaction with the target STAT3 (Table S5).

VEGFA is a growth factor that has an important role in angiogenesis, vasculogenesis and endothelial cell growth. Apigenin (TP5), luteolin (TP13), quercetin (TP14) and prunasin (TP33) showed interaction with the target VEGFA (Table S5). Hyperglycemic situations result in overexpression of VEGFA which is a critical factor in diabetic complications such as diabetic retinopathy (*Caldwelll et al., 2003*).

## CONCLUSIONS

In the last decade, network pharmacology driven omics methods play an important role to understand the role of the herbal prescriptions used as a folk medicine in various diseases. In this study, molecular mechanisms of *T. polium* in the treatment of T2DM were evaluated with the help of bioinformatics. Though there were studies on *T. polium* extract's antihyperglycemic effect via in vitro and in vivo assays, the underlying molecular mechanisms has not totally determined yet. We constructed a comprehensive dataset of compounds reported in *T. polium*. 126 compounds previously isolated or determined from the aerial parts of the plant were listed through a detailed literature search. In this view, this study serves the most detailed data on the content of the phytochemical profile of *T. polium* so far.

In the present network pharmacological analysis, insulin resistance and PI3K-AKT signaling pathway were shown to take place in the center of the mechanism of action of *T. polium*. *T. polium* is an insulin-sensitizing plant. Even though insulin resistance has an important role in the pathophysiology of T2DM, insulin resistance does not result in T2DM in all cases. It turns to T2DM with a loss in β-cell mass in pancreatic islets. The results of the present network pharmacology studies taken together with the previously reported data, suggested that *T. polium* could be a promising herb for the treatment of T2DM through ameliorating insulin resistance and enhancing β-cell mass.

### Funding
The authors received no funding for this work.

### Competing Interests
The authors declare there are no competing interests.

### Author Contributions
- Vahap Murat Kutluay and Neziha Yagmur Diker conceived and designed the experiments, performed the experiments, analyzed the data, prepared figures and/or tables, authored or reviewed drafts of the paper, and approved the final draft.

### Data Availability
The raw data is available in the Supplementary Files.

## Supplemental Information

Supplemental information for this article can be found online at http://dx.doi.org/10.7717/peerj.10111#supplemental-information.

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
