# Peer review of "Constitution of a comprehensive phytochemical profile and network pharmacology based investigation to decipher molecular mechanisms of Teucrium polium L. in the treatment of type 2 diabetes mellitus"

_PeerJ, doi:10.7717/peerj.10111_

## Round 0.1 · original submission · Major Revisions

Please disregard the comments of reviewer-1. We prefer to let the field decide on significance. As this individual had no substantive comments, this view can be ignored.

Both other reviewers recognise the potential importance of this work and have provided a detailed critique. Although this is a long list, many of the comments should be straightforward to address. However, as the list is quite extensive I have deemed this a 'major revision'.

My view, however, is that careful editing and detailed rebuttal letter should provide me with scope to make a decision on this without returning it to the reviewers.

Please note that tables are long and I agree with reviewer-3 that these could be moved to supplementary material.

Thanks for submitting this interesting study.

Reviewer 1 ·

Basic reporting

The MS is well organized, but lack innovation.

Experimental design

The experimental design and method used in this MS is out of date. As many informations from databases were not reliably, so that some of the obtained results will mistake. In a word, integrate databases in quite not enough.

Validity of the findings

As I said previous, I can't judge the fingdings is true or not even without any experimental validation.

Additional comments

An increasing phenomenon is appeared that many researchers would like to choose network pharmacology to study herbal medicies. Howerer, if there is any improvement or innovation and just follow the routine which used previous. I don't think it is a good way to promote the development of herbal drugs. At least, experimental validation is necessary.

·

Basic reporting

Diabetes is a metabolic disease, with type 2 diabetes being the most common, usually in adults. It occurs either when there is an insufficient insulin production by pancreas or when the produced insulin is not effectively utilized in the body. These disturbances result in hyperglycaemia, or raised blood sugar, which eventually leads to serious damage to many of the body’s system over time. Diabetes prevalence has dramatically risen over the years and creates concerns. Amidst this, finding new remediation or new insights to Type 2 diabetes milletus (T2DM) will be a beneficial contribution.
Authors have used bioinformatics approach to characterize the therapeutic potential of the compounds from Teucrium polium L. in the treatment of T2DM. A comprehensive phytochemical profile mined from existing literature is presented along with network-based pharmacology investigation to decipher molecular mechanisms involved.
Introduction highlights the use of the plant extracts in the treatment in various ailments. It sheds light into its bioactive compounds and the phytotherapeutic effects.
The article is well structured, and professionally written. There are some minor issues that needs to be corrected and some sentences require rephrasing for clarity. The details of which are provided along with other comments.
Overall, the article is well within the scope and it will be of interest to the readers of the journal. The article is well organized into sections and sub-sections.

Experimental design

Research question is well defined and meaningful. The authors experimental design seems appropriate for network pharmacological analysis.

Validity of the findings

Authors have mentioned, in this study they found GSK3β, GLUT4, and PDX1 to play important role in the antidiabetic effect of T. polium. But looking at the results from this study, it hard to find their role. They are not presented in any figures or table. Although authors have discussed the compounds that interact with GSK3β and GLUT4 in the discussion section, but it is not clear whether authors found it in this study by their analysis or presented them from previous literature. For PDX1 the interacting compounds are not mentioned. It will be appropriate to refer to the findings in the study, mentioning whether it is presented (table or a figure).
In conclusions, authors mention insulin resistance and PI3K-AKT signaling pathway as the center of the mechanism of action. Authors can elaborate how they finalized these pathways to be the center. Based on the network pharmacological analysis presented in this study, there are other pathways also that seem to be involved more than insulin resistance (Figure 6 and Table 3). For instance, HIF-1 signaling pathway, TNF signaling pathway, and FoxO signaling pathway.

Additional comments

Overall, the study is interesting and a useful addition to the area. However, there are certain concerns that needs be addressed to bridge the knowledge gap. A revision is recommended for improving the manuscript.

Page 2, Line 56: Authors have mentioned “Besides these key targets, with this study GSK3β, GLUT4, and PDX1 were also shown to play important role in the antidiabetic effect of T. polium”. Authors are recommended to refer in which step they identified them.
Later in discussion section (page 8 and 9), authors have mentioned the compounds that interact with GSK3β and GLUT4 but it is not clear whether authors found them in their study or mined them from previous literature. If found in this study, the corresponding figure/ table needs to be presented. Similarly, for PDX1 the information is missing, which compounds interact and how authors identified them.

Page 2, Line 60: Authors mention only 2 pathways to be the main pathways regulated by T. polium. It will appropriate to mention how they arrived at this conclusion. On the other hand, the pathway analysis (Figure 6 and Table 3) show HIF-1 signaling pathway, TNF signaling pathway, FoxO signaling pathway, and insulin signaling pathway as well.

Figure 3: The legend mentions 32 key targets but only 31 are shown in the figure. “ESR1” is missing. Check for the discrepancy. (ESR1 is present in PPI network- Figure 4)

Figure 7: The interaction of 29 compounds with 31 key targets is presented. Again, ESR1 is missing.

Other minor comments:
Page 2, Line 61-62: It will appropriate to rephrase the following sentence for clarity, “This study reveals the relationship of the compounds and targets in T. polium”. The current sentence gives an impression that targets are in T. polium whereas targets are being investigated in humans.
Page 2, Line 68: It seems there is a typing error, “…about 1,6 million deaths...”, it should be 1.6 million.
Page 2, Line 74-75: Revise the sentence for clarity to the readers, “Teucrium L. genus….”.
Page 2, Line 76-78: Revise the sentence for clarity to the readers, “In Turkish folk medicine, treatment….”.
Page 3, Line 76-86: Italicise in vivo and in vitro.
Page 3, Line 93-94: Revise the sentence for clarity to the readers, “In blood glucose concentration, ...”.
Page 3, Line 115: Acronym TCM can be described.
Page 3, Line 120: In the following sentence it will be appropriate to write “Compounds reported from T. polium were selected based on their drug-likeness properties…”.
Page 4, Line 136-138: Revise the sentences to bring more for clarity.
Page 4, Line 143: It will be appropriate to write, “ …compounds, were subjected to…”.
Page 5, Line 164: Remove extra punctuation mark after the sentence.
Page 5, Line 182: It will be appropriate to write, “The results were listed based on their p-values.”, instead of “…due to their p-values.”.
Page 6, Line 217: Authors can mention STRING 11.0 instead of String.
Page 6, Line 237: For the consistency throughout the manuscript, it will better if authors use either p-values or p values.
Page 7, Line 278: It will be appropriate to write, “..cell function which are caused by…:, since authors are referring to two defects that arise due to long-term hyperglycemia or revise the sentence accordingly to bring more clarity.
Page 8, Line 302: Italicise T. polium.
Page 8, Line 315-136: Authors mention, “Even GSK3β has not been found as a key target in our PPI network, a large …”. But authors have mentioned in the results that they found GSK3β as a target in this study. It will appropriate to correct the sentence and provide the reference to figure/ table/ supplementary data where it was found.
Page 8, Line 316-317: Italicise T. polium.
Page 9, Line 326-328: Authors mention they find interaction of the listed compounds with GLUT4 in this study. It is recommended to provide the reference to figure/ table/ supplementary data where it was found.
Page 9, Line 328 and 330: Italicise T. polium.
Page 10, Line 370: Italicise in vivo and in vitro.
Page 10, Line 369-371: Revise the following sentence for clarity, “Though there were studies...”.

References
Page 12, Line 453, 459: DOI is missing.
Page 13, Line 481: DOI is missing.
Page 13, Line 511-512: DOI can be moved to line 511.
Page 13, Line 515: DOI is missing.
Page 15, Line 557, 568: DOI is missing.
Page 15, Line 586: Proper punctuation is required.
Page 15, Line 593: DOI is missing and “-” after punctuation needs to be removed.
Page 16, Line 602: DOI is missing and reference has repeated 2020. Delete the repetition.
Page 16, Line 610: Inconsistent referencing format. Please check.
Page 16, Line 612, 619, 621, 633: DOI is missing.

Figures
Figure 1 Legend: Since the color coding of the nodes remains the same in all figures. Authors can mention in Figure 1 that the color coding scheme is consistent across figures instead of repeating the same text in Figure 3 and Figure 7.
Figure 7 Legend: The following text is redundant as in Figure 5, “Bars represent p-values…”. It can be clearly mentioned in Figure 5 that these representations are consistent in the plots instead of duplicating text.

Tables
Table 1 Legend: Authors can simply mention T. polium. In the table heading, it will be better if authors write References instead of Ref.
Table 2 Legend: It will be appropriate to write, “…29 compounds that were used in ...”.

Supplementary data
Supplemental_data_S2 and Suplemental_data_S3: The heading can be corrected to “Gene Symbol” instead of “Gene Symbole”.

Reviewer 3 ·

Basic reporting

The authors proposed an analysis to decipher molecular mechanisms of Teucrium polium L. in the treatment of type 2 diabetes mellitus. Although some vital analyses have been performed, there are some major points that need to be addressed:

English needs to be polished. The manuscript should be formatted better and some spelling and grammar should be checked carefully. For examples, some grammatical errors and typos are as follows:

- ... were put in to DAVID database ...
- Our findings suggest its use as an effective herbal drug in the treatment of T2DM and provides new insights ...
- The ‘Compound-Target’ network consist of ...
- The top 20 results due to the p-values also suggested that compounds reported from T. polium might also leads new insights for the treatment of cancer.
- ...

Experimental design

The authors should show detail on how they performed literature search i.e., which keywords or sources that they used?

The authors transformed the compounds to SMILE format using PubChem (https://pubchem.ncbi.nlm.nih.gov/) or CS Chemdraw Ultra, so in which cases/compounds they use PubChem or Chemdraw Ultra?

Why did the authors set the confidence score as high (> 0.7).

GO database or analysis has been used in previously biomedical works such as PMID: 31277574 and PMID: 31921391. Therefore, the authors should refer more works in this description to attract broader readership.

Methodology has not been described clearly and it makes hard to reproduce the results. The authors should describe it clearly.

Validity of the findings

Some figures and network analyses are redundant and necessary. For example, Fig. 1, 3, or 4 contained few information and no-sense.

Some tables are long and it is better if the authors could put them to supplementary materials.

The authors should validate the results on an unseen data.

Additional comments

No comment.

---

## Round 0.2 · accepted · Accept

Thank you for your clear and focussed response to the comments of the reviewers. I am satisfied with these and have recommended that you article now be published.